# The role of individual and community empowerment as drivers of contraceptive use among reproductive aged women in Bangladesh: Insights from multilevel analysis using BDHS 2022

**Bikash Pal** (ORCID)*, **Md. Abdus Salam Akanda**

Department of Statistics, University of Dhaka, Dhaka, Bangladesh

* bikashpal@du.ac.bd

## Abstract

This study explores the impact of individual and community-level women's empowerment on contraceptive use in Bangladesh, a country where disparities in access and utilization of modern family planning services persist. Drawing on socio-ecological theory, the research examines how both personal agency and the broader social environment interact to influence women's reproductive health decisions. Mann–Whitney U tests and chi-square tests were used for unadjusted comparisons, followed by multilevel logistic regression to account for clustering at the community level. Using data from the 2022 Bangladesh Demographic and Health Survey (BDHS), the study finds that community-level empowerment was significantly and positively associated with contraceptive use, whereas individual empowerment showed a positive but marginal association (p-value ≈ 0.10). However, community empowerment appears to have a stronger and more consistent effect than individual empowerment. Key socio-economic factors, such as age, education, and residence also significantly influence contraceptive use. The findings underscore the role of community-level empowerment in shaping women's reproductive health decisions. Community-based strategies, such as women's support groups, health volunteers, and local leadership engagement, may offer more sustainable improvements in contraceptive use than individual-focused approaches. This study adds to the growing evidence base on empowerment and reproductive health, and provides actionable program design in similar sociocultural contexts.

## Introduction

The intersection of women's empowerment and reproductive health has emerged as a pivotal area of focus in the pursuit of gender equality and sustainable development.

which permits unrestricted use, distribution, and reproduction in any medium, provided the original author and source are credited.

**Data availability statement:** The original dataset used in this study is publicly available from the Demographic and Health Surveys (DHS) Program repository at: https://dhspro-gram.com/data/available-datasets.cfm. The specific dataset extracted and used for this analysis can be accessed through the following link: https://drive.google.com/file/d/1Y-JW0yocVjU_gBZKhezU5RUOwSonZ_qKM/view?usp=sharing.

**Funding:** The author(s) received no specific funding for this work.

**Competing interests:** The authors have declared that no competing interests exist.

The Sustainable Development Goals (SDGs) 3 and 5, which aim to improve health and achieve gender equality, underscore the importance of promoting reproductive health and women's rights. Despite progress in women's education and family planning access, barriers remain in low- and middle-income countries such as Bangladesh. Inadequate access to and utilization of family planning resources has led to an estimated 14 million unintended pregnancies annually across the globe, contributing to unsafe abortions, maternal mortality, and broader socio-economic consequences for families and communities [1]. In Bangladesh, where access to modern contraception remains limited, these figures reflect a failure to meet the reproductive health needs of women, especially those in marginalized communities. Although education often increases contraceptive use [2], disparities remain in rural and conservative areas where social norms strongly influence women's choices [3]. This suggests that focusing solely on individual-level variables may overlook critical contextual influences. In particular, gender norms, cultural practices, and social attitudes within a community may play a significant role in shaping women's autonomy and ability to make informed decisions about their reproductive health. For instance, research has shown that in communities with strong patriarchal norms, where gender equality is limited, women are less likely to use modern contraceptives [4]. On the other hand, in communities where gender equality is more pronounced, women may face fewer barriers to accessing contraception and enjoy greater support from both their families and peers [5]. Furthermore, sociocultural norms are deeply rooted in patriarchal values in Bangladesh that often limit women's autonomy, especially regarding reproductive decision-making. Traditional gender roles frequently position men as the primary decision-makers, while women's reproductive choices are subject to family or societal expectations. Religious beliefs, though diverse in interpretation, can sometimes reinforce conservative views that discourage contraceptive use. Furthermore, discussing contraception publicly is often considered taboo, which creates barriers to open communication and informed choice. These cultural constraints highlight the importance of examining both personal and collective empowerment in this context.

In Bangladesh, where patriarchal structures continue to dominate social life, understanding the relationship between community-level empowerment and individual contraceptive use is critical. It is essential to recognize that a woman's ability to access family planning services and make informed reproductive choices is not only influenced by her personal level of empowerment but also by the social context in which she lives. Societal norms, community beliefs, and local attitudes toward gender roles can significantly affect women's decision-making power, including their ability to make choices about contraception [6]. Although prior studies in Bangladesh have examined women's empowerment and contraceptive use, most rely on older DHS waves, focus primarily on individual-level indicators, or do not explicitly model community-level empowerment. Moreover, few studies use PCA-based composite empowerment indices combined with multilevel modeling to disentangle individual and contextual effects. Using the most recent BDHS 2022 data, this study addresses these gaps by constructing individual and community empowerment indices and estimating their independent associations with contraceptive use within a multilevel framework.

The socio-ecological theory posits that human behavior is influenced by multiple, interacting levels of influence: individual, interpersonal, community, and societal [7]. In the context of contraceptive use, individual-level empowerment reflects personal agency and autonomy. Meanwhile, community-level empowerment, measured as the aggregate empowerment within a community or cluster, reflects the broader social norms and collective agency that can either support or constrain individual behavior. Guided by socio-ecological theory, this study conceptualizes contraceptive use as the result of interacting influences at multiple levels. At the individual level, empowerment reflects women's agency, decision-making autonomy, and rejection of gender-based violence, which can enhance confidence and negotiation within partnerships. At the community level, aggregated empowerment reflects shared gender norms, social acceptance of women's autonomy, and collective agency that shape expectations, information flows, and support for contraceptive behavior. We hypothesize that community-level empowerment exerts an independent influence on contraceptive use beyond individual empowerment by creating enabling social environments that normalize and legitimize family planning.

## Methods

### Data

This study utilized the latest Bangladesh Demographic and Health Survey (BDHS) dataset conducted in 2022. Women aged 15–49 from randomly selected households were targeted using a two-stage cluster sampling design. To ensure national representation, the survey covered 675 enumeration areas, of which 237 were from urban and 438 from rural areas. Data extraction focused on the individual record (IR file) from the BDHS dataset. A total of 20,160 households were initially selected, but the analysis was restricted to currently married women aged 15–49 with complete information on contraceptive use, empowerment indicators, and covariates. Women who were not currently married or had missing information on key variables were excluded. After sequentially removing observations with missing values, the final analytic sample consisted of 18,632 women.

### Ethics approval and consent to participate

This study is based on publicly available secondary data from the 2022 Bangladesh Demographic and Health Survey (BDHS), accessed through formal approval from the DHS Program. The original survey protocols were reviewed and approved by the Institutional Review Board (IRB) of ICF International, USA, and the Bangladesh Medical Research Council. Prior to data collection, informed consent was obtained from all participants by trained field staff. The dataset used in this study was fully anonymized and does not contain any identifiable information, ensuring the privacy and confidentiality of respondents. As the authors did not participate in the original data collection and worked only with de-identified secondary data, this study posed no additional ethical risks and involved minimal potential for researcher bias.

### Outcome variable

This study considers women's current contraceptive use as the outcome variable. The original variable, which includes multiple categories, has been simplified into a binary outcome: 'no' for non-use and 'yes' for all other categories [8].

### Independent variable

Guided by the socio-ecological theory, this study explores women's empowerment at both individual and community levels as the key independent variables. Individual empowerment reflects a woman's own capacity for decision-making and her attitudes toward gender-based violence, which captures her intrapersonal agency. Community empowerment represents the average level of empowerment across women in a cluster (enumeration area) to reflect shared norms and collective agency. This allows for examining how the social context may influence individuals beyond their personal traits.

## Individual empowerment

Women's empowerment at the individual level has been assessed through two key dimensions: participation in household decision-making and attitudes toward wife beating [9,10]. Decision-making participation was measured by asking women about their involvement in four areas: decisions regarding their healthcare, major household purchases, visits to relatives, and the use of their husband's earnings. Responses were recorded as binary (yes = 1, no = 0), where "yes" indicated that the woman had a role in the decision, either alone or jointly. Attitudes toward wife beating was assessed based on five scenarios in which a husband might be justified hitting his wife, which includes going out without permission, neglecting children, arguing, refusing to have sex, or burning food. Women who rejected all justifications were coded as 1 (empowered), while those who accepted at least one justification were coded as 0. These nine indicators were combined using Principal Component Analysis (PCA) [11,12] to generate a continuous empowerment score for each individual (Fig 1).

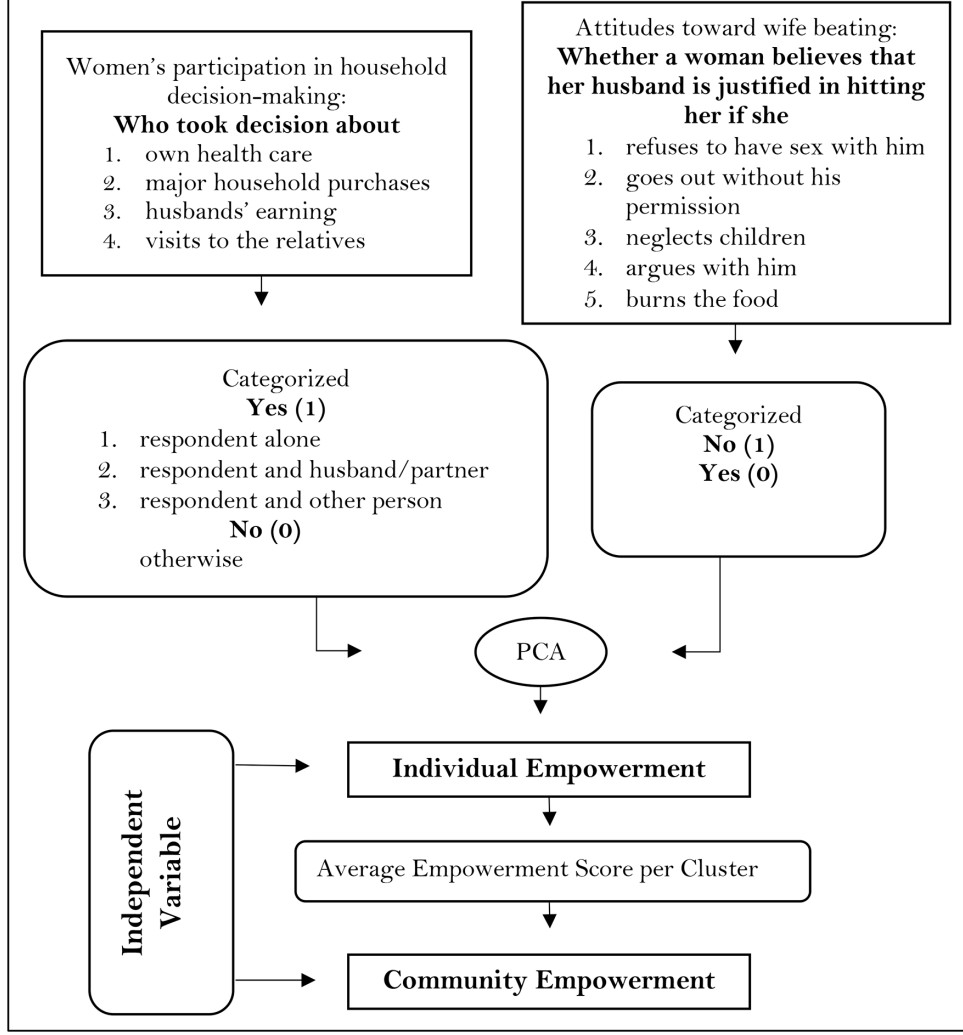

**Fig 1. Flowchart of constructing women's empowerment.**

## Community empowerment

To capture the broader social environment, a community-level empowerment variable has been created by aggregating individual empowerment scores within each cluster (enumeration area) (Fig 1). The community empowerment score represents the average empowerment level of all women within a given cluster [4]. Community-level averages were computed using data from all women in the PSU to capture the average beliefs and behavior of women in the community, rather than limiting the measure to parous women only.

## Control variable

Building upon prior research, this study incorporates several control variables that may influence the relationship between women's empowerment and current contraceptive use [13–15]. The selected variables include: Women's current age (<25, 25–35, >35), Husband's current age (<30, 30–40, >40), Women's education (No education, Primary, Secondary, Higher), Number of living children (1–2, >2), Place of residence (Rural, Urban), Working status (Yes, No), Media exposure (Yes, No), and Wealth index (Poor, Middle, Rich). The DHS wealth quintiles were collapsed into three categories: poor (poorest and poorer), middle, and rich (richer and richest) [16].

## Statistical analysis

To construct a composite measure of women's empowerment, Principal Component Analysis (PCA) was applied to derive an index score from multiple indicators. PCA is widely adopted in Demographic and Health Surveys (DHS) for constructing composite indices such as the wealth index, due to its capacity to reduce dimensionality by capturing the maximum variation from correlated variables into fewer components. Following this established practice, we used PCA to combine multiple empowerment indicators into a single score, without assuming a predefined factor structure. The principal components ($PC_m$) are calculated as:

$$PC_m = \sum_k w_{mk} X_k$$

where $w_{mk}$ represents the weight of the $k$-th variable in the $m$-th principal component, and $X_k$ are the original variables [17].

The Mann-Whitney U test has been employed to compare the distribution of empowerment scores between women who use contraceptives and those who do not [18,19]. Since empowerment scores are not normally distributed, the Mann-Whitney U test provides a non-parametric alternative to assess whether there was a significant difference between these groups. Additionally, to examine the bivariate associations between contraceptive use and various control variables, Chi-square tests have been conducted [20,21].

Given the hierarchical nature of the data, where women are nested within clusters, multilevel mixed-effects logistic regression models with random intercepts at the cluster level were fitted [22]. The multilevel logistic regression model is represented by the equation

$$In \left( \frac{p_{ij}}{1 - p_{ij}} \right) = \beta_0 + \beta_1 x_{ij1} + \beta_2 x_{ij2} + \cdots + \beta_k x_{ijk} + u_j,$$

where $p_{ij}$ is the probability of contraceptive use of the $i$th woman in the $j$th cluster, $x_{ijk}$ represents individual and community-level predictors such as women's empowerment and other socio-economic or demographic characteristics, $\beta_k$ be the regression coefficients corresponding to the $k$th independent variable, and $u_j$ be the random effect term for the $j$th cluster [20,22]. Measures of between-cluster variation were assessed using the cluster-level variance, intra-class correlation coefficient (ICC), and median odds ratio (MOR). The proportional change in variance (PCV) was computed to quantify the extent to which covariates explained between-cluster variability. Model fit was evaluated using

log-likelihood, Akaike Information Criterion (AIC), Bayesian Information Criterion (BIC), and likelihood ratio tests comparing nested models [23].

Multicollinearity among independent variables has been assessed using the Variance Inflation Factor (VIF) to identify potential correlations [24]. To further explore the association between women's empowerment and contraceptive use, visualizations have been created based on predicted probabilities and empowerment levels. All statistical analyses are performed using SPSS (version 26) and R (version 4.3.2).

## Results

Table 1 presents the results of the Mann-Whitney U test, which has been conducted to compare empowerment scores between women who use contraceptives and those who do not. The median individual empowerment score is higher among women who use contraceptives (median = 0.340, IQR = 1.300) compared to those who do not (median = 0.297, IQR = 1.310). Similarly, the community empowerment score is higher among contraceptive users (median = 0.049, IQR = 0.414) than non-users (median = 0.032, IQR = 0.402). The p-values (<0.001) in both cases indicate strong statistical evidence that contraceptive use is associated with higher levels of both individual and collective empowerment.

Table 2 presents the unadjusted association between contraceptive use and various socio-economic and demographic characteristics. The second and third columns display the row percentages of contraceptive use across different categories. All the Chi-square test p-values are found to be less than 0.001, which indicate strong statistical associations between contraceptive use and all selected variables.

Table 3 presents the results from the multilevel binary logistic regression model to examine the adjusted association between contraceptive use and key explanatory variables. The analysis accounts for both individual- and community-level factors while adjusting for socio-economic and demographic characteristics.

The results indicate that women living in more empowered communities have higher odds of using contraception (OR = 1.252, p-value = 0.004), even after adjusting for other factors. In contrast, individual-level empowerment is also positively associated with contraceptive use, though with lower statistical significance (OR = 1.028, p-value = 0.100). This indicates that while individual empowerment contributes to contraceptive decision-making, its effect is weaker compared to community-level empowerment.

Among the control variables, women aged 25–35 years have 22.6% higher odds of utilizing contraceptive methods compared to those under 25, while women older than 35 do not show a significant difference. Similarly, women married to partners aged 30–40 years are associated with 13.7% higher odds of contraceptive use, whereas no significant association is observed for partners older than 40.

Women who completed primary (OR = 1.502, p-value < 0.001), secondary (OR = 1.457, p-value < 0.001), or higher education (OR = 1.290, p-value < 0.001) being significantly more likely to use contraception compared to those with no formal education. Women with more than two living children are also more likely to use contraception (OR = 1.983, p-value < 0.001).

Residence, employment status, and media exposure also show strong associations with contraceptive use. Women residing in urban areas have 1.346 times odds of using contraception than their rural counterparts. Similarly, employed

**Table 1. Mann-Whitney U Test Results for Comparing Empowerment Scores by Contraceptive Use Among Currently Married Women Aged 15–49, BDHS 2022.**

| Contraceptive use | | Median | IQR | Effect Size | W Statistic | p-value |
|---|---|---|---|---|---|---|
| Individual empowerment | Yes | 0.340 | 1.300 | 0.042 | 37813922 | **<0.001** |
| | No | 0.297 | 1.310 | | | |
| Community empowerment | Yes | 0.049 | 0.414 | 0.043 | 37651313 | **<0.001** |
| | No | 0.032 | 0.402 | | | |

**Table 2.  Chi-square Test of Association between Contraceptive Use and Several Socio-economic Characteristics Among Currently Married Women Aged 15–49, BDHS 2022\*.**

| Variable | Contraceptive use | | p-value |
|---|---|---|---|
| | No | | Yes |
| **Women's current age** | | | <0.001 |
| <25 | 42.4 | 57.6 | |
| 25-35 | 31.7 | 68.3 | |
| >35 | 34.8 | 65.2 | |
| **Husband's current age** | | | <0.001 |
| <30 | 42.2 | 57.8 | |
| 30-40 | 33.8 | 66.2 | |
| >40 | 34.3 | 65.7 | |
| **Women's education** | | | <0.001 |
| No education | 38.2 | 61.8 | |
| Primary | 31.2 | 68.8 | |
| Secondary | 36.1 | 63.9 | |
| Higher | 38.9 | 61.1 | |
| **Number of living children** | | | <0.001 |
| 1-2 | 39.0 | 61.0 | |
| >2 | 27.6 | 72.4 | |
| **Place of residence** | | | <0.001 |
| Rural | 37.0 | 63.0 | |
| Urban | 32.8 | 67.2 | |
| **Working status** | | | <0.001 |
| No | 38.6 | 61.4 | |
| Yes | 28.4 | 71.6 | |
| **Media exposure** | | | <0.001 |
| No | 39.0 | 61.0 | |
| Yes | 33.0 | 67.0 | |
| **Wealth index** | | | <0.001 |
| Poor | 33.3 | 66.7 | |
| Middle | 35.1 | 64.9 | |
| Rich | 37.7 | 62.3 | |

\*Note that values in the second and third column represents row percentages.

women (OR = 1.422, p-value < 0.001) and/or those with media exposure (OR = 1.372, p-value < 0.001) are more likely to use contraception.

Notably, wealth index shows an inverse association as the women from middle-income or rich households have 15.2% and 31.7% lower odds, respectively, of using contraceptives compared to those from poor households. Moreover, multicollinearity was assessed using the Variance Inflation Factor (VIF), and all values were below 10, which indicates the absence of multicollinearity in the model. Additionally, the estimated cluster-level variance was 0.17, corresponding to an ICC of 5.0%, indicating that approximately five percent of the total variation in contraceptive use was attributable to differences between clusters. The median odds ratio (MOR) was 1.49, suggesting that, for two women with identical individual characteristics drawn from different clusters, the median difference in the odds of contraceptive use was 49% due solely to cluster-level factors. After adjustment for individual- and community-level covariates, the cluster-level variance decreased

**Table 3. Multilevel Binary Logistic Regression Model Estimates of the Selected Variables for Contraceptive Use in Bangladesh from BDHS 2022 Data along with Standard Error (SE), p-value and Odds Ratio (OR).**

| Variable | Estimate | SE | p-value | OR |
|---|---|---|---|---|
| **Individual empowerment** | 0.028 | 0.017 | 0.100 | 1.028 |
| **Community empowerment** | 0.225 | 0.078 | **0.004** | 1.252 |
| **Women's current age** | | | | |
| <25 | – | – | – | – |
| 25-35 | 0.204 | 0.052 | **<0.001** | 1.226 |
| >35 | −0.065 | 0.072 | 0.362 | 0.937 |
| **Husband's current age** | | | | |
| <30 | – | – | – | – |
| 30-40 | 0.129 | 0.055 | **0.019** | 1.137 |
| >40 | 0.098 | 0.072 | 0.172 | 1.104 |
| **Women's education** | | | | |
| No education | – | – | – | – |
| Primary | 0.407 | 0.057 | **<0.001** | 1.502 |
| Secondary | 0.376 | 0.057 | **<0.001** | 1.457 |
| Higher | 0.255 | 0.071 | **<0.001** | 1.290 |
| **Number of living children** | | | | |
| ≤2 | – | – | – | – |
| >2 | 0.685 | 0.043 | **<0.001** | 1.983 |
| **Place of residence** | | | | |
| Rural | – | – | – | – |
| Urban | 0.297 | 0.052 | **<0.001** | 1.346 |
| **Working status** | | | | |
| No | – | – | – | – |
| Yes | 0.352 | 0.038 | **<0.001** | 1.422 |
| **Media exposure** | | | | |
| No | – | – | – | – |
| Yes | 0.316 | 0.036 | **<0.001** | 1.372 |
| **Wealth index** | | | | |
| Poor | – | – | – | – |
| Middle | −0.165 | 0.047 | **<0.001** | 0.848 |
| Rich | −0.381 | 0.045 | **<0.001** | 0.683 |

by 11.8%, as indicated by a proportional change in variance (PCV) of 0.12, demonstrating that the included covariates explained a meaningful proportion of the between-cluster heterogeneity. Model fit improved significantly after inclusion of the explanatory variables. The full model showed a marked reduction in AIC (23243 vs. 23964) and BIC (23384 vs. 23980) compared with the null model. Likelihood ratio testing further confirmed that the final model provided a significantly better fit to the data than the null model ($\chi^2 = 753.0$, df $= 16$, p-value $< 0.001$).

To further enhance the interpretation of the regression results, a heatmap was constructed based on the odds ratios derived from the multilevel logistic regression model (Fig 2). The color gradient represents the magnitude and direction of the associations, with green shades indicating positive relationships (OR > 1) and red shades indicating negative relationships (OR < 1). The figure clearly highlights that community-level empowerment, women's education, employment, urban residence, and media exposure are positively associated with contraceptive use, whereas higher wealth status shows an

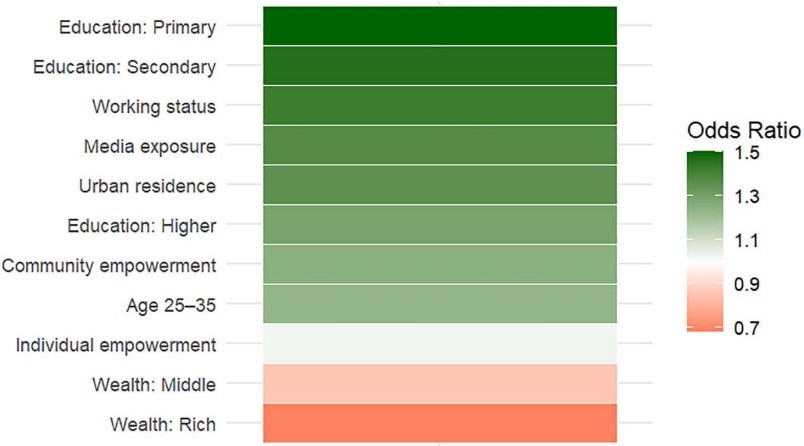

**Fig 2. Heatmap of odds ratios for contraceptive use.**

inverse relationship. This visualization provides an intuitive summary of the relative influence of each factor on contraceptive behavior.

Fig 3 illustrates the relationship between empowerment and the predicted probability of contraceptive use based on the multilevel binary logistic regression model. The x-axis represents empowerment levels, with community empowerment on the left and individual empowerment on the right, while the y-axis shows the predicted probability of contraceptive use. The scatter points represent individual observations, and the blue lines indicate the predicted probability trends with shaded confidence intervals.

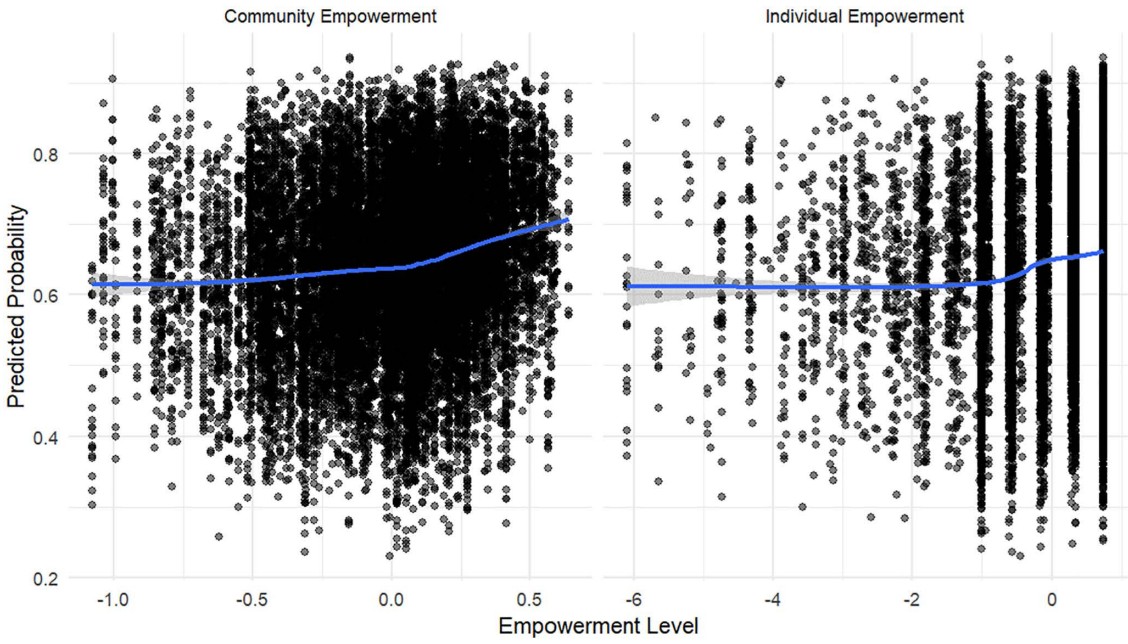

**Fig 3. Predicted probability of contraceptive use by levels of empowerment.**

The results indicate a positive association between community empowerment and contraceptive use, as the probability of contraceptive use gradually increases with higher levels of community empowerment. In contrast, the association between individual empowerment and contraceptive use appears weaker. The predicted probability remains relatively flat at lower levels of individual empowerment and only shows a slight upward trend at higher levels. Overall, the findings suggest that while both collective and individual empowerment positively influence contraceptive use, community empowerment has a stronger and more consistent effect. This aligns with the statistical results presented at Table 3, where collective empowerment was found to be highly significant (p-value = 0.004), whereas individual empowerment was significant at the 10% level (p-value = 0.100).

## Discussion

This study contributes to a growing body of research exploring the impact of women's empowerment on reproductive health decisions, with a specific focus on contraceptive use in Bangladesh. Our findings support the socio-ecological theory by demonstrating that both individual- and community-level factors significantly influence contraceptive use. Previous studies have consistently shown that empowerment, particularly gender equality, enhances open communication between partners regarding reproductive health, enhances women's access to reproductive health services, and ultimately leads to improved health outcomes [25,26]. Our analysis extends these findings by providing robust evidence from a nationally representative dataset, emphasizing the importance of both individual and community empowerment in facilitating contraceptive adoption. In our study, community empowerment exerted a stronger influence on contraceptive use than individual autonomy, which indicates the importance of shared gender norms and collective agency in shaping reproductive behavior. This finding demonstrates the importance of broader community engagement in empowering women and improving access to family planning services. In contrast, individual empowerment is positively associated with contraceptive use, but its effect is found to be weaker and less statistically significant. This suggests that while individual empowerment contributes to contraceptive decision-making, its influence is less pronounced than the influence of community factors. These findings may be explained by the fact that women's decisions are often shaped not only by personal autonomy but also by social norms, cultural values, and access to resources within their communities. Therefore, community-level empowerment initiatives, such as awareness campaigns and the strengthening of health infrastructure, may be more effective in promoting contraceptive use than individual empowerment efforts alone.

Several socio-economic and demographic characteristics have been found to significantly influence contraceptive use. Consistent with earlier research, middle-aged women show greater contraceptive uptake, likely reflecting increased reproductive decision-making power. [27,28]. Similarly, women with more than two living children are more likely to use contraceptives, which may reflect a desire to control family size once a certain number of children has been reached [29].

Our finding supports existing literature that emphasizes the positive influence of education on family planning decisions [30]. Educated women typically have better access to health services, more effective communication with their partners, and a greater understanding of reproductive health, which empowers them to make informed choices regarding contraception. Our study also finds that women residing in urban areas, those employed, and those with media exposure have significantly higher odds of using contraceptives, consistent with the view that urban settings and economic empowerment provide women with better access to family planning resources [31–33]. On the contrary, the study reveals an inverse association between wealth status and contraceptive use [34]. This unexpected finding challenges the assumption that wealth automatically correlates with increased access to family planning services. In affluent households, traditional or conservative values may persist, emphasizing lineage continuation or a preference for larger families, which can reduce contraceptive uptake despite greater accessibility. Moreover, the inverse association may reflect differences in method mix and fertility preferences. Wealthier women may prefer traditional or periodic methods not captured as "use," desire larger families, or face normative expectations regarding lineage continuation. Additionally, quality of services and provider bias may vary by socioeconomic context, and urban–wealth interactions could obscure simple linear gradients.

## Implications for policy and practice

The study findings suggest that family planning programs in Bangladesh should incorporate both individual- and community-level empowerment components. Empowerment messages and activities can be integrated into existing community health worker and family welfare assistant programs, enabling routine household visits and community sessions to address women's decision-making, spousal communication, and reproductive autonomy alongside contraceptive counseling.

Community-based women's groups and forums can be leveraged to challenge restrictive gender norms and strengthen collective agency, particularly in rural settings. In addition, engaging local leaders, including religious and elected representatives, through targeted sensitization initiatives may help legitimize contraceptive use and women's empowerment within communities. Complementary media and community communication strategies, such as radio programs and locally tailored messaging, can reinforce these norm-changing efforts. Together, these approaches can enhance the effectiveness and reach of family planning services in Bangladesh.

## Limitations and future research

While this study provides valuable insights into the relationship between women's empowerment and contraceptive use, several limitations should be acknowledged. First, the cross-sectional design prevents establishing causality; higher empowerment may lead to greater contraceptive use, or contraceptive adoption may enhance empowerment. Longitudinal data would help clarify these relationships. Second, reliance on self-reported information may introduce recall or social desirability bias, particularly in conservative contexts where women might overreport socially accepted behaviors. Third, the empowerment index was limited to available BDHS indicators and did not capture aspects such as political participation or social support. In addition, unmeasured cluster-level factors, such as local health infrastructure, NGO activities, or access to family planning services, may influence the results. Future research should apply causal modeling and mixed-methods designs to better identify mechanisms linking empowerment and contraceptive behavior. Qualitative approaches like focus groups or in-depth interviews could reveal how social norms, male partners, and community leaders shape reproductive decisions. Studies could also examine how collective empowerment mediates the relationship between education and contraceptive use.

## Conclusion

In conclusion, this study highlights the significant role of both individual and community empowerment in influencing contraceptive use among Bangladeshi women. While individual empowerment contributes to contraceptive decision-making, community empowerment has a more pronounced and consistent effect. Socio-economic and demographic factors, such as education, age, and residence, also play a critical role in shaping women's contraceptive choices. These findings provide valuable evidence for policymakers and practitioners seeking to design effective family planning programs to empower women and promote reproductive health in Bangladesh.

## Acknowledgments

We express our appreciation to the DHS (Demographic and Health Surveys) for granting us access to their dataset for our research. Furthermore, we acknowledge the National Institute of Population Research and Training (NIPORT) for conducting the BDHS, 2017–2022.

## Author contributions

**Conceptualization:** Bikash Pal, Md. Abdus Salam Akanda.

**Data curation:** Bikash Pal.

**Formal analysis:** Bikash Pal.

**Methodology:** Bikash Pal.

**Software:** Bikash Pal.

**Supervision:** Md. Abdus Salam Akanda.

**Visualization:** Bikash Pal.

**Writing – original draft:** Bikash Pal.

**Writing – review & editing:** Bikash Pal, Md. Abdus Salam Akanda.

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
