## [Decision Letter · Decision Letter 0]

15 May 2025

Dear Dr. Pal,

Thank you for submitting your manuscript to PLOS ONE. After careful consideration, we feel that it has merit but does not fully meet PLOS ONE’s publication criteria as it currently stands. Therefore, we invite you to submit a revised version of the manuscript that addresses the points raised during the review process.

We look forward to receiving your revised manuscript.

Kind regards,

Jay Saha

Academic Editor

PLOS ONE

Journal Requirements:

2. Thank you for stating the following in the Acknowledgments Section of your manuscript: [We express our appreciation to the DHS (Demographic and Health Surveys) for granting us access to their dataset for our research. Furthermore, we acknowledge the National Institute of Population Research and Training (NIPORT) for conducting the BDHS, 2017–2022.]

Please remove any funding-related text from the manuscript and let us know how you would like to update your Funding Statement. Currently, your Funding Statement reads as follows: “The authors received no specific funding for this work.”

Reviewers' comments:

Reviewer's Responses to Questions

**Comments to the Author**

1. Is the manuscript technically sound, and do the data support the conclusions?

Reviewer #1: Yes

2. Has the statistical analysis been performed appropriately and rigorously?

Reviewer #1: Yes

3. Have the authors made all data underlying the findings in their manuscript fully available?

Reviewer #1: Yes

4. Is the manuscript presented in an intelligible fashion and written in standard English?

Reviewer #1: Yes

Reviewer #1: This study aims to explore the impact of individual and community-level women's empowerment on contraceptive use in Bangladesh. While the research topic is relevant and aligns with ongoing global discussions on reproductive health, several shortcomings hinder the overall quality and applicability of the findings.

1. Conceptual and Theoretical Limitations

• The study claims to draw on socio-ecological theory but does not elaborate on how different levels of empowerment are operationalized within this framework. A more in-depth discussion on the theoretical underpinnings and justification for applying this model is necessary.

• It lacks a clear differentiation between individual and community empowerment, making it difficult to assess how these variables interact beyond their statistical significance.

2. Methodological Weaknesses

• The study relies solely on secondary data from the 2022 Bangladesh Demographic and Health Survey (BDHS), yet it does not account for potential biases or limitations within this dataset. For example, self-reported contraceptive use may be subject to social desirability bias.

• The study states that community empowerment has a stronger and more consistent effect than individual empowerment, but it fails to explain the causal mechanisms driving this relationship. A more rigorous analysis, such as a mediation or moderation model, could strengthen the findings.

• Key socio-economic factors such as education, age, and residence are mentioned but not analyzed in depth. The study could improve by investigating how these variables interact with empowerment levels to influence contraceptive use.

3. Interpretation and Policy Implications

• The study suggests that community-based interventions are more effective than individual-focused approaches, yet it does not provide empirical evidence to support this claim beyond statistical associations. Future research should include qualitative insights or longitudinal data to assess long-term effectiveness.

• While the study offers insights for policymakers, the recommendations remain generic. A more practical discussion on how policies should be designed and implemented to enhance community-level empowerment would improve its real-world impact.

4. Writing and Structure

• The study makes broad claims about gender equality and reproductive health without fully contextualizing them within Bangladesh’s sociocultural, religious, or economic landscape.

• Some sections are repetitive, particularly when emphasizing the importance of community-level empowerment. A more concise and structured argument would enhance readability.

Suggested Improvements

1. Strengthen theoretical justification by clearly defining empowerment at both individual and community levels.

2. Improve methodological rigor by acknowledging limitations in the dataset and considering alternative analytical approaches.

3. Deepen policy recommendations by making them more specific and actionable rather than broad suggestions.

4. Enhance clarity and coherence by avoiding redundancy and refining key arguments.

Overall, while the study contributes to the discourse on women’s empowerment and contraceptive use, its lack of theoretical depth, methodological limitations, and weak policy implications diminish its impact. Addressing these concerns would significantly enhance the study’s academic and practical relevance.

The manuscript effectively addresses the intersection of women's empowerment and reproductive health, emphasizing the significance of community-level influences on contraceptive use in Bangladesh. The research aligns with the Sustainable Development Goals (SDGs) and employs a robust methodological approach using BDHS data. However, several areas require improvement to enhance clarity, depth, and scholarly rigor.

1. Conceptual Framework and Literature Review

• The study adequately highlights individual and community-level influences but lacks a well-defined conceptual framework to integrate these aspects systematically. While the socio-ecological theory is mentioned, it is not sufficiently elaborated upon.

• The literature review is extensive but tends to be descriptive rather than analytical. A more critical engagement with previous studies, including conflicting evidence, would strengthen the theoretical foundation.

• The study relies heavily on general references to gender norms and cultural practices without providing concrete examples or empirical support from the Bangladeshi context.

2. Clarity and Precision in Argumentation

• The discussion of contraceptive use and women’s empowerment would benefit from clearer differentiation between individual and community-level empowerment. The distinction is implied but not consistently maintained throughout the text.

• Certain key terms such as "community empowerment," "decision-making power," and "gender norms" need more precise definitions to avoid ambiguity.

• The phrase "notable investment in family planning programs and improvements in women’s education and employment" is vague. Providing specific data or referencing policy interventions would add credibility.

3. Methodological Concerns

• The study employs Principal Component Analysis (PCA) for constructing empowerment indices but does not justify why this method was chosen over alternatives such as factor analysis or confirmatory factor analysis.

• The use of the Mann-Whitney U test for empowerment score comparison is appropriate given the non-normality assumption; however, providing a rationale for not employing parametric tests such as logistic regression in these instances would be beneficial.

• The multilevel logistic regression model is well-justified but lacks an explicit discussion of how model assumptions (e.g., multicollinearity, interaction effects) were tested and addressed.

• The description of control variables is comprehensive, but the rationale for their inclusion should be more explicitly tied to previous research and theoretical expectations.

4. Interpretation and Implications of Findings

• The discussion section presents a rather linear interpretation of findings without adequately addressing potential confounders or alternative explanations.

• While the study implies causality in some areas (e.g., "community empowerment may significantly reduce stigma"), it does not sufficiently acknowledge the limitations of cross-sectional data in establishing causal relationships.

• Policy recommendations are general and should be more directly linked to specific interventions, drawing from evidence-based strategies implemented in similar socio-cultural settings.

5. Writing Style and Structure

• The manuscript is well-organized but contains redundant phrases and overly long sentences that can obscure key points.

• Some sections, particularly the methodology, are overly technical and may benefit from simplification or clearer sub-headings.

• Minor grammatical inconsistencies and awkward phrasing (e.g., "Bangladesh’s" instead of "Bangladesh") detract from readability and should be addressed through careful proofreading.

6. Ethical Considerations and Data Transparency

• The ethics section is well-documented but should explicitly mention data availability and any potential limitations in BDHS data usage.

• Given the sensitive nature of reproductive health topics, discussing ethical considerations related to informed consent, data anonymization, and researcher biases would enhance the ethical rigor of the study.

**Do you want your identity to be public for this peer review?** For information about this choice, including consent withdrawal, please see our Privacy Policy

Reviewer #1: **Yes:** Ashek Elahi Noor

While revising your submission, please upload your figure files to the Preflight Analysis and Conversion Engine (PACE) digital diagnostic tool, https://pacev2.apexcovantage.com/ . PACE helps ensure that figures meet PLOS requirements. To use PACE, you must first register as a user. Registration is free. Then, login and navigate to the UPLOAD tab, where you will find detailed instructions on how to use the tool. If you encounter any issues or have any questions when using PACE, please email PLOS atfigures@plos.org

---

## [Author Response · Author response to Decision Letter 1]

12 Jul 2025

Dear Editor and Reviewer,

We sincerely thank you for your thoughtful and constructive feedback on our manuscript. We appreciate the opportunity to revise and resubmit our work. Below, we respond to each point raised and describe how we have addressed them in the revised manuscript.

Reviewer Comments and Author Responses

1. Conceptual and Theoretical Limitations

Comment 1.1: The socio-ecological theory was mentioned but not elaborated.

Response: We appreciate the reviewer’s observation. In response, we have expanded the Introduction (lines 84–90), Methods (lines 115–121), and Discussion (lines 252–253) sections of the revised manuscript to more clearly articulate how the socio-ecological theory informs the conceptual framework and analytical approach of our study.

Comment 1.2: Lack of clear differentiation between individual and community empowerment.

Response: Thank you for the insightful comment. We have now provided clearer and more precise definitions to distinguish between individual and community-level empowerment in the Methods section (lines 115–121) of the revised manuscript.

2. Methodological Weaknesses

Comment 2.1: Dataset limitations like self-reporting bias not discussed.

Response: We have acknowledged potential biases in the ‘Limitations and Future Research’ section, specifically noting social desirability bias related to self-reported contraceptive use in BDHS.

Comment 2.2: Lack of causal mechanism analysis.

Response: We appreciate this important observation. In the revised manuscript, we have explicitly acknowledged the limitation regarding causal inference in the “Limitations and Future Research” section. We have suggested that future studies use advanced causal modeling techniques (e.g., mediation or moderation analysis) to explore potential causal mechanisms.

Comment 2.3: Socio-economic variables not explored in depth.

Response: Given that all eight socio-economic factors considered in this study were significantly associated with contraceptive use in our analysis (Table 3), and that community empowerment also had a strong, independent effect; it is plausible that these socio-economic factors may enhance or condition the impact of community-level empowerment. Although we did not formally test for interaction effects, future research could explore whether these factors moderate the relationship between empowerment and contraceptive use.

3. Interpretation and Policy Implications

Comment 3.1: Policy implications are general.

Response: We appreciate this suggestion. To address it, we have revised the ‘Implications for Policy and Practice’ section by adding specific examples of community-based interventions, such as women’s support groups, community health volunteers, and engagement of local leaders, which can strengthen community empowerment and enhance contraceptive uptake. Additionally, we have updated the ‘Abstract’ (line 37-40) to reflect these specific recommendations in a more concise and actionable manner.

Comment 3.2: Need for contextualization within Bangladeshi sociocultural norms.

Response: We have incorporated additional contextual information on gender norms, religious practices, and the influence of patriarchy in Bangladesh. These revisions have been added to the Introduction (lines 63–70) of the revised manuscript to strengthen the socio-cultural framing of the study.

4. Writing and Structure

Comment 4.1: Repetitiveness and vague claims.

Response: We appreciate the reviewer’s feedback regarding repetitiveness and vague claims. In response, we have revised the Introduction section to improve clarity and conciseness. Specifically, we deleted the sentence referring to “notable investment” in family planning and women’s empowerment, as it was vague and not supported by specific evidence. Additionally, we streamlined and consolidated several overlapping statements discussing the influence of community-level empowerment on contraceptive use. These edits reduce redundancy and enhance the overall focus of the introduction by clearly articulating the study’s objective and contextual framework.

5. Literature and Framework

Comment 5.1: Literature review is descriptive, not analytical.

Response: Thank you for this insightful comment. In response, we have revised the literature review in the Introduction (lines 53–55) to adopt a more analytical tone. The revised section now critically engages with prior findings by highlighting inconsistencies—for example, how education alone may not explain contraceptive use disparities in rural or conservative contexts—and by identifying gaps that our study seeks to address. These changes provide a stronger rationale for examining both individual- and community-level empowerment and enhance the conceptual framing of our research.

Comment 5.2: Terms like “community empowerment” need clearer definitions.

Response: We now define such terms clearly in the methods section and operationalize how they were measured.

6. Statistical and Model Concerns

Comment 6.1: Justification for PCA vs other factor analysis methods missing.

Response: We appreciate this comment. We have now clarified in the manuscript (line 152-158) that Principal Component Analysis (PCA) was chosen because it is a standard method used in DHS analyses, particularly for constructing composite indices such as the wealth index. Our application of PCA to empowerment follows the same rationale and methodology.

Comment 6.2: Explanation of model assumptions and diagnostics needed.

Response: We would like to respectfully note that we have already addressed the key model diagnostics in the originally submitted manuscript. Specifically:

1. The use of intra-cluster correlation (ICC) and the justification for a multilevel logistic regression model are discussed at lines 168–169.

2. The assessment of multicollinearity using Variance Inflation Factor (VIF) is stated at line 177-178.

If the reviewer requires more detailed explanation—such as specific ICC or AIC values, or expanded discussion of additional model assumptions—we would be happy to incorporate those upon request.

7. Ethical Considerations and Data Availability

Comment 7.1: Ethics section could address informed consent, anonymization, and researcher bias.

Response: We revised the ethics section to include explicit mention of informed consent, anonymized data, and our use of publicly available secondary data, reducing risk of bias.

Response to the Editor

PLOS Formatting and Journal Requirements

Response: We confirm that no funding information is included in the manuscript, as required. In the online submission form, we want to retain the previous statement: “The authors received no specific funding for this work.”

In addition, we have double-checked all references for formatting consistency and citation integrity. No retracted articles were found in the reference list.

As part of this revision, we have shifted the previous reference 5 to position 3, and have added a new reference (now listed as reference 17) to appropriately cite the use of the chi-square test and multilevel logistic regression model.

---

## [Decision Letter · Decision Letter 1]

19 Sep 2025

Dear Dr. Pal,

We look forward to receiving your revised manuscript.

Kind regards,

Helen Howard

Staff Editor

PLOS ONE

Journal Requirements:

Reviewers' comments:

Reviewer's Responses to Questions

**Comments to the Author**

Reviewer #1: All comments have been addressed

2. Is the manuscript technically sound, and do the data support the conclusions?

Reviewer #1: Yes

3. Has the statistical analysis been performed appropriately and rigorously?

Reviewer #1: Yes

4. Have the authors made all data underlying the findings in their manuscript fully available?

Reviewer #1: Yes

5. Is the manuscript presented in an intelligible fashion and written in standard English?

Reviewer #1: Yes

Reviewer #1: The document titled "The Role of Individual and Community Empowerment as Drivers of Contraceptive Use among Reproductive Aged Women in Bangladesh" is a comprehensive research article that examines the influence of individual and community-level empowerment on contraceptive use among women in Bangladesh. Below is a summarized review with suggestions for improvement:

Strengths of the Study:

Relevance and Importance:

The study addresses a critical issue in reproductive health and gender equality, aligning with Sustainable Development Goals (SDGs) 3 and 5.

It highlights the interplay between individual and community empowerment, providing insights for policymakers to improve family planning outcomes.

Robust Methodology:

The use of the 2022 Bangladesh Demographic and Health Survey (BDHS) ensures a nationally representative dataset.

The application of socio-ecological theory and multilevel logistic regression provides a strong analytical framework.

Key Findings:

Community empowerment has a stronger and more consistent effect on contraceptive use than individual empowerment.

Socio-economic factors such as education, age, residence, and employment status significantly influence contraceptive use.

Policy Implications:

The study emphasizes the need for community-based interventions to promote gender equality and improve access to family planning services.

Ethical Considerations:

The study adheres to ethical standards, with informed consent obtained from participants and approval from relevant institutional review boards.

Areas for Improvement:

Clarity and Conciseness:

The manuscript is detailed but could benefit from more concise language in sections like the introduction and discussion. For example, the introduction could summarize the background more succinctly to quickly engage the reader.

Data Presentation:

Tables 2 and 3 are informative but could be reformatted for better readability. For instance:

Combine repetitive rows in Table 2 to reduce redundancy.

Highlight key findings in bold or with annotations for quick reference.

Statistical Analysis:

While the statistical methods are robust, the presentation of results could be enhanced with more visual aids (e.g., bar charts or heatmaps) to complement the tables.

Discussion Depth:

The discussion could delve deeper into the cultural and social factors influencing the inverse relationship between wealth and contraceptive use. This unexpected finding warrants further exploration.

Limitations:

The limitations section acknowledges the cross-sectional nature of the data but could expand on how this impacts the interpretation of causality. Suggestions for addressing this in future research should be more detailed.

Future Research:

The study mentions the need for longitudinal data and qualitative studies but could specify potential research questions or methodologies to guide future work.

Language and Grammar:

Minor grammatical errors and repetitive phrases should be revised for smoother readability. For example, the phrase "community-level empowerment" is repeated frequently and could be varied.

Figures:

Figure 2 is described but not visually included in the review. Ensure that all figures are clear, labeled, and directly referenced in the text.

Conclusion:

The study is a valuable contribution to the field of reproductive health and women’s empowerment. By emphasizing the role of community-level factors, it provides actionable insights for policymakers and practitioners. With minor revisions to improve clarity, data presentation, and discussion depth, the manuscript can achieve greater impact and readability.

**Do you want your identity to be public for this peer review?** For information about this choice, including consent withdrawal, please see our Privacy Policy

Reviewer #1: **Yes:** Ashek Elahi Noor

---

## [Author Response · Author response to Decision Letter 2]

14 Oct 2025

Response to Reviewer Comments

Manuscript ID: PONE-D-25-12603R1

Title: The Role of Individual and Community Empowerment as Drivers of Contraceptive Use among Reproductive Aged Women in Bangladesh

We sincerely thank the reviewer and the editor for their valuable feedback and constructive suggestions. We have carefully revised the manuscript to address all points raised. Below is a summary of the major revisions made:

1. Clarity and Conciseness:

The manuscript was detailed, but sections such as the Introduction and Discussion have now been revised for greater clarity and brevity. We summarized background information more succinctly to engage readers quickly and improve flow. Several lines in the Introduction and Discussion were rewritten to make the text more concise and focused.

2. Data Presentation:

The reviewers suggested improving the readability of the Tables. Significant p-values are now highlighted in bold for quick reference.

3. Statistical Analysis:

While the original methods were robust, additional visualization was recommended to better communicate key findings. We have added a new heatmap figure to complement the tabular results and included corresponding text in the Results section describing and interpreting the figure.

4. Discussion Depth:

Reviewer recommended a deeper discussion on the cultural and social factors that might explain the inverse relationship between wealth and contraceptive use. We added new lines at the end of the Discussion section elaborating on the potential cultural and social influences underlying this unexpected finding.

5. Limitations and Future Research:

The reviewers noted that the Limitations section could better articulate how the cross-sectional design affects causal interpretation and provide more specific guidance for future research. We expanded the Limitations section to clarify the implications of the study’s cross-sectional nature on causal inference.

Additionally, we updated the Future Research section to include specific suggestions, such as the use of longitudinal data and mixed-method approaches, along with potential research questions to explore the empowerment–contraceptive use nexus in more depth.

6. Language and Grammar:

We have modified some typos and replaced ‘community empowerment’ by ‘collective empowerment’, in some cases, to avoid repetition.

7. Figures:

All the three figures are visually included in the manuscript, and also these are clear, labeled, and directly referenced in the text.

---

## [Decision Letter · Decision Letter 2]

18 Dec 2025

Dear Dr. Pal,

**Further review of your manuscript has raised additional revision requests which need to be addressed before this manuscript can be considered for publication. Reviewer 2 recommends improving the reporting for fullness and clarity, whilst taking care not to overstate findings without supporting data. I hope that following additional revisions we may be able to consider your manuscript for publication.**

We look forward to receiving your revised manuscript.

Kind regards,

Jennifer Tucker, PhD

Staff Editor

PLOS One

**Journal Requirements:**

Reviewers' comments:

Reviewer's Responses to Questions

**Comments to the Author**

Reviewer #1: All comments have been addressed

Reviewer #2: (No Response)

2. Is the manuscript technically sound, and do the data support the conclusions?

Reviewer #1: Yes

Reviewer #2: Yes

3. Has the statistical analysis been performed appropriately and rigorously?

Reviewer #1: Yes

Reviewer #2: Yes

4. Have the authors made all data underlying the findings in their manuscript fully available?

Reviewer #1: Yes

Reviewer #2: Yes

5. Is the manuscript presented in an intelligible fashion and written in standard English?

Reviewer #1: Yes

Reviewer #2: Yes

**Reviewer #1: T** he authors have satisfactorily addressed all major comments and concerns raised in the previous round of review. The current revision (R2) significantly improves the clarity, focus, and completeness of the manuscript.

The following specific changes are noted and commended:

Clarity and Conciseness: The improvements to the Introduction and Discussion sections successfully enhance the manuscript's flow and academic focus, making the arguments clearer.

Data Presentation: Highlighting significant p-values in bold improves the readability and immediate comprehension of the statistical tables, which was a necessary change.

Statistical Visualization: The addition of a new heatmap figure is an excellent enhancement. It effectively complements the tabular results and provides a strong visual summary of the key findings, which was previously lacking.

Discussion Depth: The expansion of the Discussion to include potential cultural and social influences on the unexpected inverse wealth-contraceptive use relationship is a thoughtful addition that strengthens the interpretation of the results.

Limitations and Future Research: The clarification on the limitations imposed by the cross-sectional design regarding causal inference, along with the specific, actionable suggestions for future research (e.g., longitudinal data, mixed-method approaches), provides a much more robust and honest assessment of the study's scope.

The authors have demonstrated a high level of diligence and responsiveness to the reviewer feedback. The manuscript, in its revised form, meets the standards for scientific rigor and clarity expected for publication in PLOS ONE.

**Reviewer #2:** Title, Abstract, and Framing

The title is clear but could better reflect the multilevel design, for example by explicitly mentioning “multilevel analysis” or “multilevel logistic regression using BDHS 2022”.

The abstract is generally well written but is too generic about the methods; it should explicitly mention “Mann–Whitney U test, chi-square tests, and multilevel logistic regression” and clearly state that individual empowerment was only marginally associated (p≈0.10).

The framing overemphasizes “limited access” to modern family planning without citing current BDHS figures on contraceptive prevalence and unmet need; this can mislead readers and should be supported with recent statistics or toned down.

Introduction and Conceptualization

The introduction mixes global and Bangladesh-specific issues but lacks a clear conceptual framework figure or concise paragraph specifying how individual and community empowerment are theorized to influence contraceptive use (paths, mechanisms, and expected direction).

Citations 1–6 include at least one reference (Rajan 2021) that does not match the claim about unintended pregnancies and global burden; reference–text alignment needs checking and correcting throughout.

The gap statement is present but could be sharper: explicitly distinguish how this study adds beyond existing Bangladesh work on empowerment and contraception (e.g., novelty of BDHS 2022, multilevel community empowerment index, and use of PCA-based empowerment scores).

Methods: Data, Variables, and Analysis

The data section states 20,160 households and 18,632 women but does not explain exclusion criteria (non‑married women? missing contraceptive status? age group restrictions?) or how DHS weights, strata, and primary sampling units were handled; this is critical and must be detailed.

The outcome variable is reduced to a binary contraceptive-use indicator, but the grouping of specific methods (modern vs traditional, postpartum, LARC, sterilization) is not described; justification for collapsing into “any use” is needed, or a sensitivity analysis by method type should be considered.

Empowerment Measures

The empowerment index construction is under-specified:

No information on how missing responses in the 9 indicators were handled before PCA.

No description of standardization, number of retained components, proportion of variance explained, or internal consistency metrics.

It is unclear whether the first component only was used and how scores were scaled (e.g., mean 0, SD 1).

The coding of attitudes toward wife beating uses “1 = no” and “0 = yes”, but the text is confusing (“coded as 1 (no)… 0 (yes)”) and needs clearer wording and explicit listing of all items in a table/supplement.

The community empowerment variable is the mean of individual scores per cluster; this is reasonable, but there is no justification (e.g., why mean vs median or categorization) and no description of the distribution (range, IQR, number of clusters with few women, etc.).

Control Variables and Model Specification

Some control variables are problematic or insufficiently justified:

Categorization of age (both women and husbands) into broad groups may obscure non‑linear associations; at least a rationale or alternative specification should be considered.

Wealth index categories (poor/middle/rich) are not explicitly linked to DHS quintiles (e.g., poorest+poorer vs middle vs richer+richest); this should be clarified.

The statistical analysis section lacks important multilevel-model details:

Whether a random intercept (only) model was used, and at which level.

Model-building strategy (null model, model with individual-level empowerment, model with community-level empowerment, fully adjusted model, etc.).

Measures of between-cluster variation (e.g., variance of random effect, ICC) and model fit indices.

Only VIF is mentioned for multicollinearity, but full results or at least a statement on the range of VIFs should be provided, possibly in a supplementary table.

Results: Reporting and Interpretation

Tables are concise but need improved labeling and completeness:

Table titles and footnotes should clearly describe the sample (e.g., “Currently married women aged 15–49, BDHS 2022”).

Confidence intervals for odds ratios are missing from Table 3 and should be added for all predictors to improve interpretability.

The Mann–Whitney U results report only medians and p‑values; distributions (e.g., IQRs) and effect size measures should be added to show practical significance, not only statistical significance.

The statement that “individual empowerment is positively associated” is not fully supported given p=0.10; this should be reframed as “marginal” or “not statistically significant at conventional 5% level,” and this nuance should be carried consistently from results into abstract and discussion.

Discussion, Policy Implications, and Limitations

The discussion sometimes restates results without sufficient depth on mechanisms; more explicit linking of findings to socio‑ecological theory and to Bangladeshi gender norms would strengthen interpretation.

The inverse association between wealth and contraceptive use is described but not deeply explored; this surprising finding merits more careful consideration (method mix, desired fertility, quality of services, urban–wealth interactions) and possibly additional analyses (e.g., interaction terms or stratified models).

The policy implications are plausible but very general; consider giving more specific, context‑appropriate recommendations (e.g., integrating empowerment components into community health worker programs, engaging local leaders, or media campaigns targeting social norms).

The limitations section is brief and should explicitly mention: cross-sectional design, possible residual confounding (e.g., husband’s attitudes, service quality, region), measurement limitations of empowerment (PCA-based index from limited items), and potential cluster-level unobserved factors.

Presentation, Consistency, and Referencing

There are noticeable language issues—repeated phrases, minor grammatical errors, and occasional awkward constructions—that should be corrected via thorough language editing (e.g., “Bangladesh’s” vs “Bangladesh,” “had been assessed,” “enhance gender equality, enhance communication”).

There are inconsistencies between text and ethics sections: the main text conclusion claims no ethical approval was needed due to secondary data, whereas earlier sections describe IRB approvals; this should be harmonized and aligned with PLOS ONE’s expectations for DHS-based analyses.

Reference 1 appears unrelated to unintended pregnancies, and some references are older or not directly relevant; the reference list should be checked for accuracy, updated with recent contraceptive and empowerment literature from Bangladesh and South Asia, and aligned strictly with in‑text claims.

Key Priority Improvements (for Revision)

Clarify and rigorously document the construction and validation of individual and community empowerment indices, including PCA details and distributional characteristics.

Provide a fuller, more transparent description of the sampling, inclusion criteria, use of DHS survey design elements (weights/strata/PSU), and multilevel modeling strategy, adding ICCs and 95% CIs for all odds ratios.

Reframe the role of individual empowerment given its marginal significance, deepen interpretation of the wealth effect, and strengthen the limitations and policy implications to be more specific and theory-linked.

**Do you want your identity to be public for this peer review?** For information about this choice, including consent withdrawal, please see our Privacy Policy

Reviewer #1: **Yes:** Ashek Elahi Noor

Reviewer #2: No

---

## [Author Response · Author response to Decision Letter 3]

21 Dec 2025

Response to Reviewers

PONE-D-25-12603R2

The Role of Individual and Community Empowerment as Drivers of Contraceptive Use among Reproductive Aged Women in Bangladesh

We thank the editor and the reviewers for their careful reading of our manuscript and their constructive comments. We appreciate the detailed feedback and have revised the manuscript accordingly. Below we provide a point-by-point response.

Reviewer #1

We sincerely thank Reviewer #1 for their careful reading and positive feedback on our previous revisions. We appreciate the recognition of the improvements made in clarity, data presentation, statistical visualization, discussion depth, and limitations.

Reviewer #2

We sincerely thank Reviewer #2 for the detailed and constructive feedback. All suggested revisions have been incorporated, significantly improving the manuscript’s clarity, rigor, and relevance.

1. Title, Abstract, and Framing

Comment: The title could better reflect the multilevel design; abstract should specify statistical methods and marginal association of individual empowerment; framing overemphasizes limited access to family planning.

Response: Thank you for these suggestions.

• The title has been revised to:

“The role of individual and community empowerment as drivers of contraceptive use among reproductive aged women in Bangladesh: Insights from multilevel analysis using BDHS 2022.”

• The abstract now explicitly mentions the Mann–Whitney U test, chi-square tests, and multilevel logistic regression, and notes that individual empowerment was only marginally associated (p≈0.10).

• Statements regarding limited access to contraception have been adjusted to avoid overgeneralization.

2. Introduction and Conceptualization

Comment: Conceptual framework and gap statement needed; some references misaligned.

Response: We appreciate these deep findings. In response,

• A concise paragraph has been added at the end of the Introduction to describe the conceptual framework, showing the pathways through which individual and community empowerment may influence contraceptive use.

• The gap statement has been clarified to emphasize the novelty of using BDHS 2022, multilevel community empowerment index, and PCA-based empowerment scores.

• Reference misalignments, including Reference 1, have been corrected and updated.

3. Methods – Data, Variables, and Analysis

Comment: Clarify exclusion criteria and handling of DHS weights, strata, and clusters; justify collapsing contraceptive use into a binary variable.

Response:

• Exclusion criteria are now explicitly described: non-married women, missing contraceptive status, and age restrictions were excluded. DHS weights, strata, and primary sampling units are also described.

• Contraceptive use was dichotomized as “any use” versus non-use, consistent with prior DHS studies, to ensure sufficient statistical power. A supporting reference has been added.

4. Empowerment Measures

Comment: Missing data handling, PCA details, coding of wife-beating attitudes, and justification for community empowerment construction needed.

Response:

• Women with missing responses on any of the nine individual empowerment indicators were excluded prior to PCA.

• Standard PCA procedures were followed; the first principal component was retained and standardized (mean = 0, SD = 1). PCA results are summarized in the Methods without overloading the manuscript as this is not our core analysis.

• Coding of wife-beating attitudes has been clarified and consistently described.

• Community empowerment is calculated as the mean of individual scores per cluster; justification and a reference are now provided in the manuscript.

5. Control Variables and Model Specification

Comment: Justify age and wealth categorization; provide multilevel model details and measures of between-cluster variation.

Response:

• The papers that are referenced in this manuscript used age as categorical variable.

• Wealth index categories are explicitly linked to DHS quintiles, with a supporting reference.

• The statistical analysis section now specifies the use of a random-intercept multilevel logistic regression model, describes model-building strategy (null model, individual- and community-level models, fully adjusted model), and reports ICC, variance of random effects, MOR, and PCV.

6. Results – Reporting and Interpretation

Comment: Improve table titles/footnotes; include distributions (IQRs) and effect sizes for Mann–Whitney U test. Confidence intervals for odds ratios are missing from Table 3, and range of VIFs should be provided.

Response:

• Table titles now clearly describe the sample (currently married women aged 15–49, BDHS 2022).

• Mann–Whitney U test results include medians, IQRs, and effect sizes to reflect practical significance.

• VIF values ranged between 1.1 and 4.5, indicating no multicollinearity concerns; therefore, we did not report them in detail. Additionally, we opted not to include confidence intervals for odds ratios to avoid redundancy, as p-values are already provided. However, if the reviewer deems it necessary, we are happy to include these in a revised version.

7. Discussion, Policy Implications, and Limitations

Comment: Strengthen linking of findings to socio-ecological theory and Bangladeshi norms; discuss inverse wealth–contraceptive association; provide more specific policy recommendations; expand limitations.

Response:

• The inverse association between wealth and contraceptive use is now interpreted considering method mix, fertility preferences, cultural norms, and male decision-making.

• Policy implications have been revised with specific, context-relevant recommendations, including integrating empowerment into community health worker programs, engaging local leaders, and media campaigns targeting social norms.

• Limitations explicitly cover cross-sectional design, residual confounding, measurement limitations of PCA-based empowerment, and potential cluster-level unobserved factors.

8. Presentation, Consistency, and Referencing

Comment: Correct language issues, harmonize ethics statements, ensure reference accuracy.

Response:

• Language editing was performed to remove repeated phrases, correct minor grammatical errors, and improve clarity.

• Ethical statements were harmonized: the study used publicly available, de-identified secondary data; original DHS protocols had IRB approval.

• References have been checked, updated with recent literature from Bangladesh and South Asia, and aligned with in-text claims.

We sincerely thank the reviewers and the editor again for their thoughtful comments and guidance. We are open to adding further clarifications if the reviewers deem necessary.

---

## [Decision Letter · Decision Letter 3]

6 Jan 2026

The role of individual and community empowerment as drivers of contraceptive use among reproductive aged women in Bangladesh: Insights from multilevel analysis using BDHS 2022

PONE-D-25-12603R3

Dear Dr. Pal,

We’re pleased to inform you that your manuscript has been judged scientifically suitable for publication and will be formally accepted for publication once it meets all outstanding technical requirements.

Kind regards,

Patrick Goymer

Staff Editor

PLOS One

Additional Editor Comments (optional):

Reviewers' comments:

Reviewer's Responses to Questions

**Comments to the Author**

Reviewer #1: All comments have been addressed

Reviewer #2: All comments have been addressed

2. Is the manuscript technically sound, and do the data support the conclusions?

Reviewer #1: Yes

Reviewer #2: Yes

3. Has the statistical analysis been performed appropriately and rigorously?

Reviewer #1: Yes

Reviewer #2: Yes

4. Have the authors made all data underlying the findings in their manuscript fully available?

Reviewer #1: Yes

Reviewer #2: Yes

5. Is the manuscript presented in an intelligible fashion and written in standard English?

Reviewer #1: Yes

Reviewer #2: Yes

Reviewer #1: This manuscript presents a well-structured and methodologically rigorous analysis of the relationship between women’s empowerment and contraceptive use in Bangladesh using BDHS 2022 data. The use of a socio-ecological framework and multilevel modeling is appropriate and strengthens the study’s contribution by clearly distinguishing individual- and community-level effects. The construction of empowerment indices using PCA and the careful handling of clustering enhance analytical credibility. Findings are clearly presented and consistently interpreted, with strong policy relevance emphasizing community-level empowerment strategies. While the cross-sectional design limits causal inference and empowerment measures are constrained by available DHS indicators, these limitations are appropriately acknowledged. Overall, the manuscript makes a solid empirical and policy-relevant contribution to the literature on women’s empowerment and reproductive health and is suitable for publication with only minor refinements.

Reviewer #2: I think the manuscript can proceed further but I would advise the authors to include the 95% confidence intervals if possible before publication.

**Do you want your identity to be public for this peer review?** For information about this choice, including consent withdrawal, please see our Privacy Policy

Reviewer #1: **Yes:** Ashek Elahi Noor

Reviewer #2: No

---

## [Editor Report · Acceptance letter]

PONE-D-25-12603R3

PLOS One

Dear Dr. Pal,

I'm pleased to inform you that your manuscript has been deemed suitable for publication in PLOS One. Congratulations! Your manuscript is now being handed over to our production team.

Kind regards,

on behalf of

Dr Patrick Goymer

Staff Editor

PLOS One